# NMT-Obfuscator Attack: Ignore a sentence in translation with only one word

**Sahar Sadrizadeh**
EPFL, Lausanne, Switzerland
sahar.sadrizadeh@epfl.ch

**César Descalzo**
EPFL, Lausanne, Switzerland
cesar.descalzo2@gmail.com

**Ljiljana Dolamic**
Armasuisse S+T, Thun, Switzerland
ljiljana.dolamic@ar.admin.ch

**Pascal Frossard**
EPFL, Lausanne, Switzerland
pascal.frossard@epfl.ch

## Abstract

Neural Machine Translation systems are used in diverse applications due to their impressive performance. However, recent studies have shown that these systems are vulnerable to carefully crafted small perturbations to their inputs, known as adversarial attacks. In this paper, we propose a new type of adversarial attack against NMT models. In this attack, we find a word to be added between two sentences such that the second sentence is ignored and not translated by the NMT model. The word added between the two sentences is such that the whole adversarial text is natural in the source language. This type of attack can be harmful in practical scenarios since the attacker can hide malicious information in the automatic translation made by the target NMT model. Our experiments show that different NMT models and translation tasks are vulnerable to this type of attack. Our attack can successfully force the NMT models to ignore the second part of the input in the translation for more than 50% of all cases while being able to maintain low perplexity for the whole input.

## 1 Introduction

Neural Machine Translation models have been deployed in many different applications in recent years due to their significant performance [Bahdanau et al., 2015, Vaswani et al., 2017]. It has been shown that these models are susceptible to adversarial attacks despite their great performance. In adversarial attacks, the input sentence is perturbed carefully to steer the output of the target model in a specific way. Even when the adversarial sentence is natural and preserves the semantics of the input sentence, the target model can be confused and generate a wrong translation.

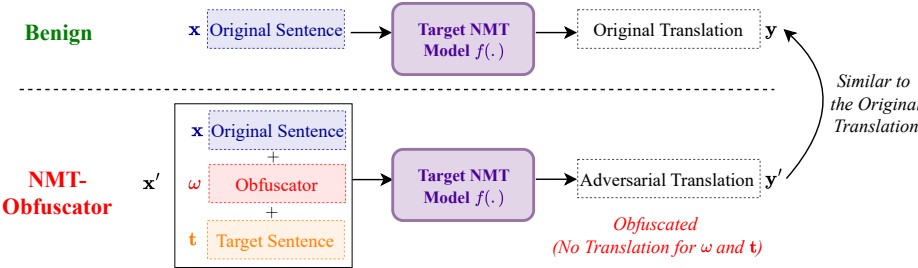

Figure 1: Block diagram of NMT-Obfuscator attack.

NeurIPS 2024 Safe Generative AI Workshop

Table 1: Illustration of an NMT-Obfuscator adversarial attack against DeepL (En-Fr). By adding the obfuscator word *"this"* between the input sentence and the target sentence, the NMT model is confused and does not translate the target sentence.

| | Input Text (red=obfuscator, *orange*=target sentence) | En-to-Fr NMT (blue=English meaning) |
|---|---|---|
| Org. | horse meat has been detected in beef food products sold by two small British low-cost distribution retailers. | De la viande de cheval a été détectée dans des produits alimentaires à base de viande de bœuf vendus par deux petits détaillants britanniques à bas prix. (Horse meat has been detected in beef food products sold by two small UK low-cost retailers.) |
| Adv. | horse meat has been detected in beef food products sold by two small British low-cost distribution retailers. this *news were actually a marketing campaign.* | De la viande de cheval a été détectée dans des produits alimentaires à base de viande de bœuf vendus par deux petits détaillants britanniques de la distribution à bas prix. (Horse meat has been detected in beef food products sold by two small UK low-cost retailers.) |

The unreliable performance of these models can pose risks in security-important applications. Hence, recent works have studied adversarial attacks against Neural Machine Translation (NMT) models and have proposed methods to make these models more robust [Ebrahimi et al., 2018a, Michel et al., 2019, Wallace et al., 2020, Cheng et al., 2020, Zhang et al., 2021, Sadrizadeh et al., 2023a].

NMT models convert a sequence in the source language to a sequence in the target language. Since the output of the model is a sequence as opposed to a single output like classifier models, different types of adversarial attacks against NMT models exist based on the adversary's objective. In untargeted attacks, which are the most common type of attacks, the adversarial sentence preserves the meaning in the source language, while it forces the target NMT model to generate a wrong translation [Ebrahimi et al., 2018a, Cheng et al., 2019, Michel et al., 2019, Niu et al., 2020, Zou et al., 2020, Sadrizadeh et al., 2023a]. Targeted attacks, on the other hand, push or mute predefined keywords from the translation while they preserve the meaning in the source language [Ebrahimi et al., 2018a, Cheng et al., 2020, Wallace et al., 2020, Sadrizadeh et al., 2023b, 2024].

On another note, the adversary may aim to hide a message when translated by the target NMT model. In other words, the adversarial attack may force the target NMT model not to translate part of the input. This type of attack can be more dangerous in comparison to simply reducing the translation quality (untargeted attacks) or pushing/muting words (targeted attacks). In this paper, we propose a new type of attack against NMT models. Given an input sentence, the adversary aims to find a word to join the input sentence and a target sentence such that when this concatenation is translated by the target model, the translation will be indistinguishable from the original translation. The overall block diagram of our attack, called NMT-Obfuscator, is depicted in Figure 1. Also, an example of our attack against DeepL API is presented in Table 1. This type of attack can be harmful in practical scenarios since the attacker can obfuscate malicious information in the automatic translation. Or they can hide part of the input in the translation that is crucial for the complete meaning of the sentence, which then can utterly change the message of the sentence as presented in the example of Table 1. The adversarial text satisfies two constraints. First, the final input, which consists of the input sentence, the obfuscator word, and the target sentence, forms a natural text for the reader in the source language.[1] Second, the target sentence is not translated by the target NMT model, and thus, the adversarial translation is highly similar to the translation of the original input sentence. Since the final adversarial text in the source language is natural and grammatically correct, the adversary can plausibly deny the intentional usage of such examples, and it would be potentially challenging to detect these attacks.

In order to generate this type of adversarial examples, we assume a white-box scenario and use gradient projection to solve our proposed discrete optimization problem. Our optimization problem consists of negative log-likelihood minimization loss between the adversarial translation and the original translation. We also use a pre-trained language model in the projection step of the algorithm to choose the best obfuscator word so that the final adversarial text is natural. To the best of our knowledge, the closest attack in the literature to our attack is the work in [Wallace et al., 2020].

---

[1]In practical scenarios, this constraint is necessary to ensure that the input text is stealthy and is not flagged automatically in the presence of a detector.

However, in this work, the phrase added between the input and target sentences is incoherent and easily perceptible. Our extensive experiments show that our attack strategy yields high-quality adversarial examples.

The rest of this paper is structured as follows. Section 2 discusses other adversarial attacks against NMT models. In Section 3, we present the problem formulation and then describe our attack algorithm. We evaluate our attack against different NMT models and discuss the results in Section 4. Finally, Section 5 concludes the paper.

## 2    Related Works

As opposed to adversarial attacks in computer vision, textual attacks are more challenging. Since the input data is discrete, it is not trivial to use gradient-based methods to generate such attacks and define the imperceptibility of adversarial perturbations.

Most of the attacks in the literature have focused on text classifiers. Some of the early attacks consider character manipulation and show that Natural Language Processing (NLP) models are not robust [Ebrahimi et al., 2018b, Gao et al., 2018, Pruthi et al., 2019]. Many other attacks perturb the input sentence at the word level and constrain the perturbations to preserve semantics. Due to the discrete nature of textual data, most of the attacks are based on word-replacements. They choose important words in the input sentence and substitute them with meaning-preserving candidates [Alzantot et al., 2018, Ren et al., 2019, Zang et al., 2020, Jin et al., 2020, Maheshwary et al., 2021, Ye et al., 2022]. Some of the attacks use gradient-based optimization methods to generate stronger attacks [Guo et al., 2021, Sadrizadeh et al., 2022, Yuan et al., 2023].

Another important task in NLP is machine translation. Since the output of the model is a sentence rather than a class, the adversary can have various objectives. In untargeted attacks, which are more common, the adversarial sentence preserves the meaning in the source language while the translation quality is reduced in the target language [Ebrahimi et al., 2018a, Cheng et al., 2019, Michel et al., 2019, Niu et al., 2020, Zou et al., 2020, Sadrizadeh et al., 2023a]. Particularly, Cheng et al. [2019] tries to reduce the translation quality by replacing random words in the input sentence with suggestions from a masked language model. Michel et al. [2019] and Zhang et al. [2021] replace important words with their neighbours in the embedding space to fool the target NMT model. On the other hand, Cheng et al. [2020] and Sadrizadeh et al. [2023a] use optimization to generate semantic-preserving adversarial sentences.

There are other types of adversarial attacks against NMT models. In targeted attacks, the adversary aims to push or mute predefined keywords from the translation [Ebrahimi et al., 2018a, Cheng et al., 2020, Wallace et al., 2020, Sadrizadeh et al., 2023b]. Wallace et al. [2020] also introduces universal attacks, where one adversarial phrase can be appended to any input and fool the target NMT model. Moreover, Sadrizadeh et al. [2024] defines a new type of attack, where the class of the output translation, which reflects the whole meaning of the sentence, is targeted.

In this paper, we propose another type of attack against NMT models. In this attack, the adversary aims to find an obfuscator word to insert between the input sentence and the target sentence such that the target sentence is not translated by the NMT model. This attack can be dangerous since the attacker can hide a message or change the meaning of the sentence (by not translating a good portion of it) in the target language. We should note that the closest attack in the literature is suffix-dropper in [Wallace et al., 2020], which aims to find a universal snippet of text, which is usually incomprehensible, to append to any input sentence such that whatever comes after it is ignored by the NMT model. Their attack replaces the trigger token with the token whose embedding minimizes the first-order Taylor approximation of the adversarial loss. In contrast to this work, we constrain the obfuscator word to be chosen such that the final adversarial text is natural. This is important since it makes the attack more stealthy and less detectable in the source language. Moreover, our attack is not universal and we find the obfuscator word for each pair of input and target sentences. Finally, we extensively evaluate the vulnerability of NMT models to our attack. However, Wallace et al. [2020] manually computes the success rate for 100 sentences.

# 3   NMT-Obfuscator Attack

In this section, we first present the problem formulation for our attack. Afterwards, we introduce the attack algorithm for NMT-Obfuscator.

## 3.1   Problem Formulation

In this section, we provide the problem formulation of our attack against NMT models. We denote the source and target language domains with $\mathcal{X}$ and $\mathcal{Y}$, respectively. The target NMT model is a mapping $f : \mathcal{X} \to \mathcal{Y}$ that translates the input sequence of tokens $\mathbf{x} = x_1 x_2 ... x_n \in \mathcal{X}$ to the output sequence $\mathbf{y} = y_1 y_2 ... y_m$. The NMT model is trained such that the likelihood of the output translation to be the ground-truth translation is maximized.

Given a target sentence $\mathbf{t} \in \mathcal{X}$, which is potentially harmful, the adversary is looking for an obfuscator token $\omega$ to insert between the input sentence and the target one. Therefore, the final adversarial input $\mathbf{x}'$ is the concatenation of the input sentence, the obfuscator, and the target sentence: $\mathbf{x}' := \mathbf{x} \,||\, \omega \,||\, \mathbf{t}$. While the adversarial input should be a natural and correct text, its translation by the target model should not include the translation of the target sentence nor the obfuscator token. Hence, the translation of the adversarial text has to be similar to the translation of the input sentence: $f(\mathbf{x}') = f(\mathbf{x})$.

In order to find the token $\omega$ that goes between the input and target sentences, the adversary needs to minimize the cross-entropy loss between the translation of the adversarial text by the NMT model and the original translation $\mathbf{y} = f(\mathbf{x})$.

$$\mathcal{L}_{Adv}(\omega) = \mathcal{L}_f(\mathbf{x} \,||\, \omega \,||\, \mathbf{t}, \mathbf{y}), \tag{1}$$

where $\mathcal{L}_f$ is the loss function of the model when the input is the adversarial example $\mathbf{x} \,||\, \omega \,||\, \mathbf{t}$ and the output translation has to be the original translation $\mathbf{y}$.

The final adversarial input should be natural. In order to impose this condition, we can use the loss of a pre-trained language model as a rough measure for the fluency of the final adversarial text. The loss of a causal language model, denoted by $\mathcal{L}_{LM}$, is the negative log-probability normalized to the sentence length. Therefore, the adversary should solve the following optimization problem:

$$\mathbf{x}' \leftarrow \underset{\omega \in \mathcal{V}}{\arg \min} \, \mathcal{L}_{Adv}(\omega) \quad s.t. \quad \mathcal{L}_{LM}(\mathbf{x} \,||\, \omega \,||\, \mathbf{t}) < \beta, \tag{2}$$

where $\mathcal{V}$ is the source language vocabulary set. Also, hyper-parameter $\beta$ determines how natural the adversarial text is.

## 3.2   Attack Algorithm

We now explain the algorithm of our attack to generate adversarial texts that satisfy the constraints discussed in the previous section. The proposed optimization problem (2) is discrete since the token $\omega$ that the adversary is looking for should be in the source language vocabulary set. In order to solve this optimization problem, we propose to use gradient projection in the embedding space of the target NMT model. As is common in NMT models, the discrete tokens of the input are first transformed into continuous embedding vectors before being fed to the model. We denote this transformation that maps the input tokens to their corresponding embedding vectors by emb(.). We can find the gradients of the adversarial loss in the embedding space of the target NMT model and then project the updated embedding vector to the nearest valid token.

The pseudo-code of our attack is presented in Algorithm 1. In more detail, we first convert the input sentence and the target sentence into their continuous representation. We also initialize the obfuscator $\omega$ with a random token from the vocabulary set $\mathcal{V}$. Afterwards, we consider the proposed optimization problem (2) in the embedding space of the target NMT model. In each iteration of the algorithm, first, we find the gradients of $\mathcal{L}_{adv}$ with respect to the embedding of $\omega$ and update the embedding vector in the opposite direction of the gradient. Then, we have to project the updated embedding vector $\mathbf{e}_\omega$ into the most similar valid one. We use cosine similarity between the embedding representations to approximate the similarity and find the most similar token in the vocabulary. Since we want the adversarial text $\mathbf{x} \,||\, \omega \,||\, \mathbf{t}$ to be natural, instead of finding the most similar valid token, we find $k$

---
**Algorithm 1** NMT-Obfuscator
---
**Input:**
    $f(.)$: Target NMT model, $\mathcal{V}$: Vocabulary set, $\mathbf{x}$: Input sentence, $\mathbf{y}$: Original translation of $\mathbf{x}$,
    $\mathbf{t}$: Target sentence, $N$: Maximum No. of iterations, $k$: No. of neighbours for projection,
    $\gamma$: Step size, $\alpha$: Distance threshold
**Output:**
    $\mathbf{x}' = \mathbf{x} \,\|\, \omega \,\|\, \mathbf{t}$: Generated adversarial text
**initialization:**
    $\mathbf{e_x} \leftarrow \text{emb}(\mathbf{x}), \quad \mathbf{e_t} \leftarrow \text{emb}(\mathbf{t}), \quad \mathbf{e}_\omega \leftarrow \text{emb}(t) : t \in_R \mathcal{V}, \quad itr \leftarrow 0,$
**while** $itr < N$ **do**
    $itr \leftarrow itr + 1$
    **Step 1:** Gradient descent in the continuous embedding space:
    $\mathbf{e}_\omega \leftarrow \mathbf{e}_\omega - \gamma . \nabla_{\mathbf{e}_\omega} \mathcal{L}_f(\mathbf{e_x} \,\|\, \mathbf{e}_\omega \,\|\, \mathbf{e_t}, \mathbf{y})$
    **Step 2:** Find $k$ nearest valid tokens in the embedding space:
    $W \leftarrow \underset{W \subset \mathcal{V}, |W|=k}{\arg\max} \; \sum_{w \in W} \frac{\text{emb}(w)^\top \mathbf{e}_\omega}{\|\text{emb}(w)\|_2 . \|\mathbf{e}_\omega\|_2}$
    **Step 3:** Find the token that minimizes the perplexity:
    $\omega \leftarrow \underset{w \in W}{\arg\min} \, \mathcal{L}_{LM}(\mathbf{x} \,\|\, w \,\|\, \mathbf{t})$
    **if** $\text{Dist}(f(\mathbf{x} \,\|\, \omega \,\|\, \mathbf{t}), \mathbf{y}) \leq \alpha$ **then**
        **return** $\mathbf{x}' = \mathbf{x} \,\|\, \omega \,\|\, \mathbf{t}$ **(adversarial attack is successful)**
    **end if**
**end while**
**return** (no obfuscator was found)
---

nearest tokens as follows:

$$\underset{W \subset \mathcal{V}, |W|=k}{\arg\max} \; \sum_{w \in W} \frac{\text{emb}(w)^\top \mathbf{e}_\omega}{\|\text{emb}(w)\|_2 . \|\mathbf{e}_\omega\|_2}. \tag{3}$$

The set $W$ contains $k$ tokens in the vocabulary whose embeddings are most similar to the updated $\mathbf{e}_\omega$. Amongst these $k$ candidate tokens, we choose the one that results in minimum loss of a pre-trained language model, $\mathcal{L}_{LM}$, to ensure that the adversarial text is fluent.

We perform these three steps iteratively until the distance between the original translation $\mathbf{y}$ and the translation of the adversarial text $f(\mathbf{x} \,\|\, \omega \,\|\, \mathbf{t})$ is less than a threshold. We use Levenshtein distance to measure the distance between the two translations since we want the two translations to be as similar as possible.

## 4 Experiments

In this section, we present the setup of our experiments and evaluate our attack against different translations tasks.[2]

### 4.1 Experimental Setup

In order to evaluate the effectiveness of our attack, we use the test set of WMT14 [Bojar et al., 2014] which is widely used for the evaluation of translation tasks, and we focus on English-to-French (En-Fr) and English-to-German tasks (En-De). We attack the HuggingFace implementations of Marian NMT models [Junczys-Dowmunt et al., 2018] and mBART50 multilingual NMT model [Tang et al., 2020]. Some statistics of these datasets alongside the translation quality of Marian NMT and mBART50 on them are reported in Table 2. Additionally, we use the GPT-2 language model [Radford et al., 2019] as the pre-trained LM in our attack algorithm. We compare our attack performance with that of Suffix-Dropper from [Wallace et al., 2020], which is the only attack in the literature with the same threat model.[3]

---

[2]Our source code is available at `https://github.com/sssadrizadeh/NMT_Obfuscator`.

[3]To make this attack consistent with our work, we slightly adapted their method. In more detail, their proposed attack is considered to be universal. However, to make a fair comparison, since in our threat model, we

In order to find the minimizer of the proposed optimization problem (2), we set the following hyper-parameters unless explicitly stated: we use Adam optimizer with a learning rate of $\gamma = 0.04$. Moreover, we perform at most $N = 100$ iterations of the attack algorithm. In the projection step of the algorithm, we find $k = 20$ most similar tokens. Finally, for the stopping criteria, we set the Levenshtein distance threshold at $\alpha = 5$.

Table 2: Some statistics of the evaluation datasets.

| Dataset | Average Length | #Test Samples | Marian NMT | | mBART50 | |
|---|---|---|---|---|---|---|
| | | | BLEU | chrF | BLEU | chrF |
| En-Fr WMT14 | 27 | 3003 | 39.88 | 64.94 | 36.17 | 62.66 |
| En-De WMT14 | 26 | 3003 | 27.72 | 58.50 | 25.66 | 57.02 |

For evaluation, we report several performance metrics: **Attack Success Rate (ASR)**, which measures the success rate of adversarial attacks. As stated in Section 3, we define the adversarial example as successful if the Levenshtein distance of the adversarial translation is no more than $\alpha$ tokens away from the original translation. **BLEU score** [Post, 2018] and **BERTScore** [Zhang et al., 2019] computed between the original and adversarial translations. These two metrics confirm that the target sentence is not translated by the target NMT model. Finally, the **Perplexity score (Perp.)** of the adversarial example measured by the GPT-2 (large) model. This metric is the most crucial one, as it shows that our attack successfully reduces the influence of the obfuscator token $\omega$ on the fluency of the overall adversarial text.

## 4.2 Performance Evaluation

We evaluate NMT-Obfuscator in comparison to the Suffix-Dropper (SD) attack from [Wallace et al., 2020] against different NMT models and translation tasks. We choose the target sentences as presented in Table 3 to perform the attack. These sentences were selected based on their potentially negative connotations, which could then be appended to the sentences in the dataset with an obfuscator word.

Table 3: Target sentences that are used in the attack.

| ID | Target Sentence |
|---|---|
| $i$ | *it's all fake* |
| $ii$ | *I planted the bomb* |
| $iii$ | *the secret code is HX532* |
| $iv$ | *the revolution will begin at 6pm* |
| $v$ | *the password is 1234, please do not share it with anyone* |

Table 4 shows the performance of these attacks. Overall, the attack is highly successful in forcing the NMT model not to translate the target sentences. Moreover, the BLEU score and BERTScore are near perfect in all cases, which confirms that the translation of the adversarial text (input sentence + obfuscator + target sentence) is very similar to the translation of the input sentence. In comparison to the baseline, we can see that both attacks have competitive success rates. However, the perplexity of the adversarial text created by our attack is much lower than that of the baseline in all cases. This means that the obfuscating token has a minimal influence on the overall sentence, making the adversarial examples more stealthy and less detectable.

Table 5 shows an adversarial example generated by NMT-Obfuscator against Marian NMT (En-Fr). We can see that by adding the obfuscator word, the NMT model is fooled and does not translate the target sentence. Moreover, the overall adversarial text is natural.

## 4.3 Analysis

According to the results of Table 4, it seems that the success rate generally decreases with the increase in the length of the target sentence. This can be due to the fact that the NMT model is less likely to ignore longer texts. To further investigate this phenomenon, we consider a fixed sentence: *"The main entrance is guarded by a security guard who is armed, I will wait for you outside the building"*. We then extract the following target sentences from it: "The main entrance", "The main

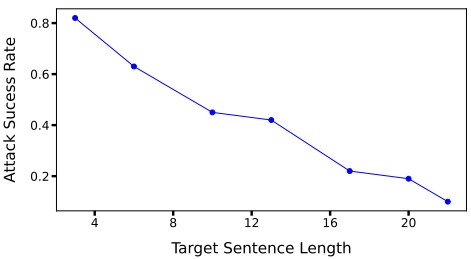

Figure 2: Effect of the target sentence length.

find the obfuscator word for each input sentence, we also run their attack for each sentence separately. We also consider the same success criteria for both attacks.

Table 4: Performance of adversarial attacks against different NMT models.

| ID | Method | Marian (En-Fr) | | | | Marian (En-De) | | | | mBART50 (En-Fr) | | | |
|----|--------|------|-------|---------|-------|------|-------|---------|-------|------|-------|---------|-------|
| | | ASR↑ | BLEU↑ | BERTSc.↑ | Perp.↓ | ASR↑ | BLEU↑ | BERTSc.↑ | Perp.↓ | ASR↑ | BLEU↑ | BERTSc.↑ | Perp.↓ |
| *i* | ours | 0.92 | 0.96 | 0.99 | 94.84 | 0.70 | 0.98 | 0.99 | 93.77 | 0.78 | 0.98 | 0.99 | 96.77 |
| | SD | 0.92 | 0.94 | 0.99 | 194.32 | 0.70 | 0.99 | 0.97 | 170.70 | 0.80 | 0.95 | 0.96 | 189.75 |
| *ii* | ours | 0.94 | 0.95 | 0.99 | 112.59 | 0.71 | 0.99 | 0.99 | 110.93 | 0.78 | 0.99 | 0.99 | 115.06 |
| | SD | 0.94 | 0.98 | 0.99 | 256.96 | 0.76 | 0.99 | 0.99 | 201.40 | 0.79 | 0.98 | 0.99 | 212.38 |
| *iii* | ours | 0.53 | 0.96 | 0.98 | 122.72 | 0.45 | 0.99 | 0.98 | 112.4 | 0.64 | 0.98 | 0.98 | 115.71 |
| | SD | 0.55 | 0.95 | 0.99 | 163.10 | 0.48 | 0.98 | 0.99 | 186.51 | 0.63 | 0.97 | 0.97 | 196.35 |
| *iv* | ours | 0.81 | 0.96 | 0.98 | 98.29 | 0.50 | 0.96 | 1.00 | 103.81 | 0.76 | 0.98 | 1.00 | 102.69 |
| | SD | 0.80 | 0.98 | 0.99 | 200.53 | 0.53 | 0.97 | 0.99 | 148.99 | 0.74 | 0.98 | 0.99 | 186.77 |
| *v* | ours | 0.58 | 0.96 | 0.99 | 96.37 | 0.14 | 0.99 | 1.00 | 43.29 | 0.58 | 0.98 | 0.99 | 54.39 |
| | SD | 0.56 | 0.96 | 0.99 | 153.19 | 0.13 | 0.97 | 1.00 | 102.16 | 0.61 | 0.99 | 0.99 | 150.01 |
| Avg. | ours | **0.76** | 0.96 | 0.99 | **104.96** | 0.5 | 0.98 | 0.99 | **92.84** | 0.71 | **0.98** | **0.99** | **96.92** |
| | SD | 0.75 | 0.96 | 0.99 | 193.62 | **0.52** | 0.98 | 0.99 | 161.95 | 0.71 | 0.97 | 0.98 | 187.05 |

Table 5: An adversarial example against Marian NMT (En-Fr) (red=obfuscator, *orange*=target sentence).

| Sentence | Text |
|----------|------|
| Org. | Earlier this evening online server trackers noted the NSA's website had been down for at least six hours, and the site continues to be inaccessible for some users. |
| Org. Trans. | Plus tôt ce soir, les traqueurs de serveurs en ligne ont noté que le site Web de la NSA était en panne depuis au moins six heures, et le site continue d'être inaccessible pour certains utilisateurs. |
| Adv. Ours | Earlier this evening online server trackers noted the NSA's website had been down for at least six hours, and the site continues to be inaccessible for some users. Although *the password is 1234, do not share it with anyone.* |
| Trans. | Plus tôt ce soir, les traqueurs de serveur en ligne ont noté que le site Web de la NSA était en panne depuis au moins six heures, et le site continue d'être inaccessible pour certains utilisateurs. |

entrance is guarded", "The main entrance is guarded by a security guard", and so forth. Figure 2 shows the success rate of our attack against Marian NMT (En-Fr) for target sentences with different lengths (number of tokens). As expected, it is harder to successfully drop the target sentence in the translation when it is longer.

We also analyze the effect of the number of iterations on our attack performance. We attack Marian NMT (En-Fr) when the target sentences are the ones from Table 3 and compute the average success rate. The results are presented in Figure 3. These results demonstrate that the success rate increases noticeably with the number of iterations until around 100 iterations, after which the success rate stabilizes.

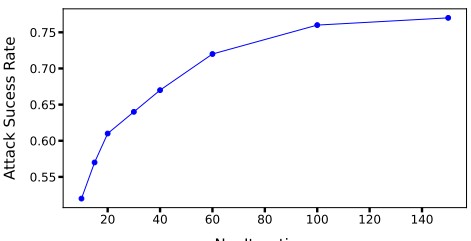

Figure 3: Effect of the number of iterations.

## 5 Conclusion

In this paper, we proposed NMT-Obfuscator, which is a new type of attack against NMT models. In this attack, the adversary aims to find an obfuscator word such that when inserted between the original sentence and the target sentence, the NMT model is fooled and does not translate the target sentence. In real-life applications, this type of attack can be very harmful since the adversary can hide a malicious message in the translated text or cause the translation to have a totally different meaning since parts of the input are not translated. In order to generate such attacks, we proposed an optimization problem and solved it using gradient projection in the embedding space of the NMT model. We also used a pre-trained language model to ensure that the final adversarial text has low perplexity. Our experimental results demonstrate that our attack is highly effective for different target sentences and against different NMT models and translation tasks.

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
