# OpenReview forum: "NMT-Obfuscator Attack: Ignore a sentence in translation with only one word"
_NeurIPS.cc/2024/Workshop/SafeGenAi — SafeGenAi Poster_

### Official Review · Reviewer_WDHy · 2024-10-09
**Comment on NMT-Obfuscator for NMT attack**

**Rating:** 6
**Confidence:** 4

**Review:**

This paper proposes an interesting adversarial attack against NMT models by adding a word between two sentences so that the model ignores the second sentence. The authors show the effectiveness of this attack using the WMT14 test set on state-of-the-art NMT models.

Strength:
1. The overall writing is well structured and easy to follow.
2. The proposed attack has a clear mathematical formulation and is well-explained.
3. The evaluation is comprehensive, and the authors provide very detailed performance metrics (ASR, BLEU, BERTScore, Perplexity) to evaluate the attack's success.

Weakness:
1. The paper lacks sufficient discussion on the set up of the experiment tasks and explanation of design choices.
2. The experiments are limited to English-to-French and English-to-German tasks. It would be interesting to have more experiments on more language pairs to see the generalizability of the NMT-Obfuscator.
3. The comparison with only the baseline method (Suffix-Dropper) may not provide a comprehensive view on whether this attack still outperforms other existing adversarial attacks in NMT.
4. Minor issue: it would be helpful to elaborate more in the abstract and conclusion of the different NMT models and success rate rather than describing in words.